# Abnormal vasculature reduces overlap between drugs and oxygen in a tumour computational model: Implications for therapeutic efficacy

Romain Enjalbert[1], Jakub Köry[2†], Timm Krüger[3☺*], Miguel O. Bernabeu[1☺*]

**1** Centre for Medical Informatics, Usher Institute, The University of Edinburgh, Edinburgh, United Kingdom, **2** School of Mathematics and Statistics, University of Glasgow, Glasgow, United Kingdom, **3** School of Engineering, Institute for Multiscale Thermofluidics, The University of Edinburgh, Edinburgh, United Kingdom

† Deceased May 2024.
☺ Equally contributing senior authors.
* timm.krueger@ed.ac.uk (TK); miguel.bernabeu@ed.ac.uk (MOB)

## Abstract

The tumour microvasculature is abnormal, and as a consequence oxygen and drug transport to the tumour tissue is impaired. The abnormal microvasculature contributes to tumour tissue hypoxia, as well as to varying drug penetration depth in the tumour. Many anti-cancer treatments require the presence of oxygen to be fully efficacious, however the question of how well oxygen concentration overlaps with drug concentration is not elucidated, which could compromise the therapeutic effect of these drugs. In this work we use a computational model of blood flow and oxygen transport, and develop a model for an oxygen-dependent drug, T-DM1, to study the overlap of oxygen and drug concentration in tumour tissue, where we model both compressed and uncompressed vessels in the tumour. Our results show that, due to the compressed vessels present in tumours, areas of sufficient oxygen concentration for a drug to function overlap poorly with areas of sufficient drug concentration, covering 28% of the tumour tissue, compared to 82% in healthy tissue. The reduction in drug and oxygen overlap is due to the altered red blood cell dynamics through the abnormal microvasculature, and indicates that drug transport to tumours should not be considered independently of oxygen transport in cases where the drug requires oxygen to function.

## Author summary

In tumours, abnormal vascular structures compromise blood flow and, as a consequence, tumour tissue is hypoxic and drug delivery is poor. In addition, some drugs require the presence of oxygen for maximum therapeutic efficacy. While oxygen and drugs are both delivered to the tissue by the blood vessels, they are

**Data availability statement:** The code to generate the microvascular networks is available at

https://github.com/thierry3000/tumorcode. The code to perform the blood flow simulations is available at

https: //github.com/Romain-Enjalbert/poiseuille_ solver/tree/compression_branch. The code to perform the oxygen and drug transport simulations is available at https://github.com/jmsgrogan/ MicrovesselChaste. All the results from simulations used in this work have been deposited in Edinburgh DataShare (https://doi.org/10.7488/ds/7967).

**Funding:** M.O.B. gratefully acknowledges funding from: Fondation Leducq Transatlantic Network of Excellence (17 CVD 03); EPSRC grant no. EP/X025705/1; British Heart Foundation and The Alan Turing Institute Cardiovascular Data Science Award (C-10180357); Diabetes UK (20/0006221); Fight for Sight (5137/5138); the SCONe projects funded by Chief Scientist Office, Edinburgh & Lothians Health Foundation, Sight Scotland, the Royal College of Surgeons of Edinburgh, the RS Macdonald Charitable Trust, and Fight For Sight. The funders had no role in study design, data collection and analysis, decision to publish, or preparation of the manuscript.

**Competing interests:** The authors have declared that no competing interests exist.

transported via different mechanisms. This work investigates, through mathematical modelling of blood flow and drug transport in a model microvascular network, how the overlap of oxygen and drugs is impaired by abnormal vascular structures in tumours. The findings show that, when vessels are compressed by tumour tissue, the different modes of transport of oxygen and drugs lead to a reduced overlap of drugs and oxygen in the tumour tissue, compromising the therapeutic efficacy of the oxygen-dependant modelled drug. Therefore, for oxygen-dependent drugs, optimising for this overlap should guide the design of future therapeutic strategies.

## Introduction

The abnormal tumour microenvironment (TME) is a hallmark of cancer [1]. The TME impairs drug delivery to tumour tissue and increases tumour tissue hypoxia, both barriers to the clinical efficacy of drugs [2–4].

Pharmacokinetics is an important aspect of the clinical efficacy of a drug [5,6]. Drugs are first transported through the blood stream and then extravasate to the tumour tissue [2,7–9]. Following extravasation, the advection, diffusion and reaction kinetics of the drug determine its transport to tumour tissue [5,6]. The TME can act as a barrier to drug delivery to the tissue and therefore to its efficacy [5,10]. However, the presence of a drug can be insufficient to fully determine a drugs efficacy, as recent studies have shown that the cytotoxicity of some drugs may also be reduced by the absence of oxygen [4,11–14]. As an example, the efficacy of trastuzumab-emtansine (T-DM1), a drug used to treat HER2+ breast cancer, is reduced under hypoxic conditions due to caveolin-1 translocation [11], making the transport of oxygen to tumour tissue of particular relevance for the efficacy of such drugs.

Oxygen travels bound to red blood cells through microvascular networks prior to being delivered to the tissue [15]. In microvascular networks, red blood cells partition unevenly at microvascular bifurcations [16]. Previous work has shown that in networks with tumour vascular abnormalities [17], such as reduced inter-bifurcation distances [18] and vessel compression [19,20], the partitioning of red blood cells at bifurcations is abnormal, leading to a high heterogeneity of discharge haematocrit (flowrate fraction of red blood cells in blood) [18–21]. As a consequence of haematocrit heterogeneity, blood vessels are inefficient at delivering oxygen and tumour tissue can be hypoxic [18,22–25]. In addition, the tumour microenvironment can have avascular areas that lead to hypoxic regions due to the limited diffusion distance of oxygen [26,27].

Treatment strategies reprogramming the tumour microvasculature have been developed with the aim to improve tumour oxygenation and drug delivery [28]. These vascular normalisation therapies modulate tumour microvascular properties, such as vascular density or permeability, followed by changes in drug delivery and tissue oxygenation depending on dosage and timing [29]. In addition, drugs inhibiting

collagen synthesis can decompress vessels, improving perfusion, oxygenation and drug delivery [22,30]. The ability to reprogramme the microvasculature asks the question of how microvascular abnormality, and its modulation, has an effect on drug and oxygen overlap from a mechanistic point of view.

In cases where a drug requires oxygen to function, the question of the overlap of drug and oxygen concentration for full therapeutic effect needs to be considered. Since oxygen and drugs are transported via different mechanisms, we hypothesise that, in the abnormal tumour microvasculature, areas of high oxygen concentration can overlap poorly with areas of high drug concentration, and that this hypothesis is relevant given the oxygen requirements of some drugs for them to be efficacious. We investigate this hypothesis in a computational model of a tumour with tumour induced compressed vessels,[31] blood flow [19], oxygen transport [18] and T-DM1 transport [7], where the computational model is used as an exemplar to illustrate how the different mechanisms of transport lead to poor overlap between oxygen and drug concentration in the tissue.

Our results show that, due to the abnormal vascular structures present in tumours (vessel compression in our computational model), areas of sufficient oxygen concentration for a drug to function overlap poorly with areas of sufficient drug concentration. This effect is mostly due to the poor oxygen transport to the tumour tissue, and indicates that drug transport should not be considered independently of oxygen transport in cases where the drug requires oxygen to function. These findings provide a theoretical underpinning for therapeutic approaches aimed at enhancing tumour oxygenation in order to improve anticancer drug efficacy.

## Methods

### Core compressed and core uncompressed models

To test our hypothesis, we generate a core compressed and a core uncompressed model to compare the two. In both models, the same microvascular network is used, illustrated in Fig 1A. Reviews of computational models to generate microvascular networks and angiogenesis are available [32,33]. We used the Tumorcode software [34], as it is open source, and the networks have been shown to recapitulate key properties matching real network samples (such as vessel size and Murray's law), see [35]. The code employs a set of rules to generate a microvascular network on a mesh, see [35] for details. The settings we use to generate the network are the default two-dimensional network generator with a single inlet and a single outlet in a 5000 $\mu m$ by 5000 $\mu m$ square domain. The network is generated as a healthy network in the code, where we later computationally introduce compression to the vessels in order to study the effect of tumour vascular compression independently of other vascular abnormalities. Additionally, we prune the dead-end vessels of the network, as shown in Fig A in S1 Text, as they would have no flow. We separate the microvascular network into periphery and core vessels, shown in Fig 1A, where core vessels are defined as having both ends of the vessel within 1000 $\mu m$ of the centre of the network [19], the remaining vessels are periphery vessels. In the core compressed model, the core vessels are compressed but not the periphery ones, while in the core uncompressed model none of the vessels are compressed.

### Blood flow

In this work, we treat blood flow as a one-dimensional continuous fluid, imposing Poiseuille's law at every vessel segment [16]. In addition, we use existing models to implement the Fåhraeus effect, Fåhraeus-Lindqvist effect, and phase separation at bifurcations in our treatment of blood flow [36–38]. In addition, we consider some blood vessels to be compressed, and in those cases treat vessel-cross sections as elliptical (with an aspect ratio of 4.26 [31]) and use a modified phase-separation model for red blood cell partitioning at bifurcations [19]. The networks were chosen to have a single inlet and a single outlet, facilitating the choice of boundary conditions as they do not affect the distribution of discharge haematocrit in the network, therefore an arbitrary pressure value is imposed at the inlet, and a 0 value at the outlet. Unless stated otherwise, the inlet discharge haematocrit to the network is 30% [39]. S1 Text contains the full details of the blood flow model.

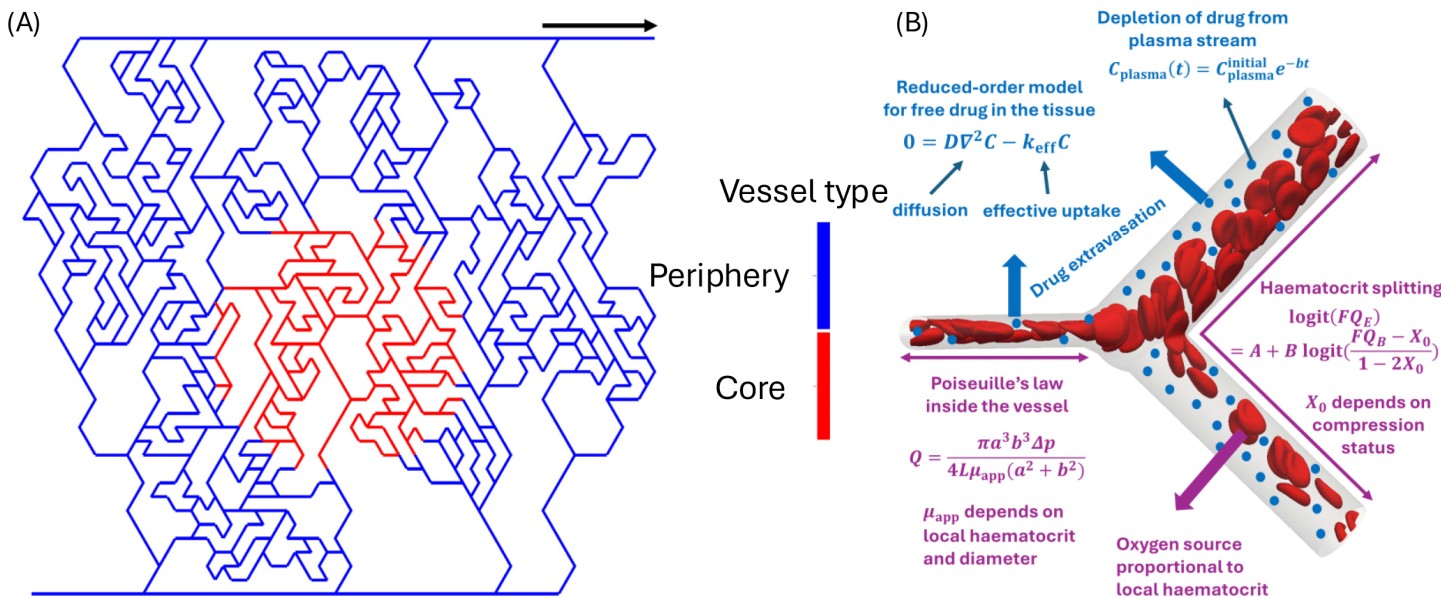

**Fig 1**. **(A) Illustration of microvascular network generated with Tumorcode [34], see methods for more details.** The black arrows indicate the inlet, at the bottom left, and the outlet, at the top right. In red are the core vessels (compressed in the core compressed model) and in blue are the periphery vessels. (B) Illustration of a diverging vessel bifurcation showing key equations of the presented model at the microscale, the red blood cells are for illustration purposes as they are not individually resolved in our model. The equations and their variables are defined in Methods section and S1 Text.

## Oxygen and drug transport

Vascular networks are embedded inside a rectangular domain representing the tissue through which oxygen and T-DM1 diffuse. Both chemical species are uptaken by the tissue and are supplied by vessels acting as line sources with strength proportional to local vessel diameter thus perfusing the tissue.

The oxygen transport problem is governed by the same equation and parameters as described in Supplementary material to [18] and the numerical simulations are performed in Microvessel Chaste [40]. We employ a simplified oxygen transport model without oxygen saturation dynamics, such as in [18], as haematocrit is more influent to determine oxygen concentration in the tissue [41]. Here, by focussing on the long-time behaviour we adopt a quasi-steady state approximation [42]. Mathematically, the problem thus reduces to a steady reaction-diffusion equation

$$D_{O_2} \nabla^2 C_{O_2} + \pi d_l \gamma \left( \frac{\beta_{ref}}{0.45} H_l - C_{O_2} \right) \delta_{network} - \kappa C_{O_2} = 0, \tag{1}$$

where $D_{O_2}$ is the diffusion coefficient for oxygen in the tissue, $C_{O_2}$ is the oxygen concentration in the tissue, $\gamma$ is the vessel permeability, $d_l$ is the local vessel diameter, $H_l$ is the local vessel discharge haematocrit, $\beta_{ref}$ relates the discharge haematocrit in the vessel to an oxygen concentration in the vessel, $\kappa$ is the rate at which oxygen is consumed by the cells, and $\delta_{newtork}$ is a representation of the vessel network via a collection of Dirac delta function sources [18]. The majority of the governing parameters are taken from [43], while the haematocrit term is taken from the blood flow simulations described in the Blood Flow section.

For the tissue dynamics of T-DM1, one must typically distinguish between three distinct phases of the drug – free, bound and internalized – and model transitions between these phases. Our starting point is the model from [7]. Free drug (concentration $C$) diffuses in the tissue with a diffusion constant $D$ and is supplied by the vessels at a rate proportional

to the plasma concentration $C_{\text{plasma}}$ (which decays in time due to depletion of drug from the blood stream, more details to follow in this section). Simultaneously, the free drug binds to receptors embedded in cell membranes (supplied at a fixed concentration $C_r$ for simplicity) to form bound complexes (concentration $B$) with a rate constant $k_{\text{on}}$ and unbinds from these complexes at a rate $k_{\text{off}}$. Drug in bound complexes is thus immobilized (cannot further penetrate the tissue) and internalized (concentration $I$) at a rate $k_{\text{int}}$. As trastuzumab is a monoclonal antibody, we assume no binding to blood cells or plasma proteins occurs ($f_{\text{free}} = 1$ holds for the free-drug fraction from [5]). The full dimensional model for T-DM1 kinetics then reads

$$\frac{\partial C}{\partial t} = D\nabla^2 C + \pi d_l P\left(C_{\text{plasma}} - \frac{C}{\varepsilon}\right)\delta_{\text{network}} - k_{\text{on}}\frac{CC_r}{\varepsilon} + k_{\text{off}}B$$
$$\frac{\partial B}{\partial t} = k_{\text{on}}\frac{CC_r}{\varepsilon} - k_{\text{off}}B - k_{\text{int}}B \qquad (2)$$
$$\frac{\partial I}{\partial t} = k_{\text{int}}B,$$

where $d_l$ is the local vessel diameter, $\varepsilon$ is the effective void fraction, $P$ the vessel permeability and $\delta_{\text{network}}$ is a representation of the vessel network via a collection of Dirac delta function sources [7,18]. Using data from [44] we found (see S1 Text) that the depletion of T-DM1 from the blood stream can be modelled via an exponential function from data from [44]

$$C_{\text{plasma}}(t) = C_{\text{plasma}}^{\text{initial}}e^{-bt}, \qquad \text{where} \quad C_{\text{plasma}}^{\text{initial}} = 810 \text{ nM} \quad \text{and} \quad b = 0.2105 \text{ day}^{-1}. \qquad (3)$$

Eq (2) with plasma concentration depleting according to Eq (3) features six physical processes - diffusion, extravasation, plasma depletion, binding, unbinding and internalization - and each of these proceeds at an associated timescale. In our network, local vessel diameters $d_l$ and inter-vessel distances $h_l$ are in the ranges 15–110 and 100–1000 $\mu$m, respectively. Using default parameters (as summarized in Table A in S1 Text), we can then estimate the values and ranges of the 6 timescales present in our simulations as follows:

$$2.36 \text{ day} < t_{\text{diff}} = \frac{h_l^2}{D} < 236 \text{ days} \qquad 0.12 \text{ day} < t_{\text{ev}} = \frac{h_l^2}{\pi d_l P} < 87.7 \text{ day} \qquad t_{\text{pd}} = 1/b = 4.75 \text{ day}$$

$$t_{\text{on}} = \frac{\varepsilon}{C_r k_{\text{on}}} = 7.68 \times 10^{-4} \text{ day} \qquad t_{\text{off}} = \frac{1}{k_{\text{off}}} = 7.82 \times 10^{-5} \text{ day} \qquad t_{\text{int}} = \frac{1}{k_{\text{int}}} = 1.32 \times 10^{-2} \text{ day}.$$

We see that the equilibration of diffusion, extravasation and plasma depletion processes happens much slower (days to months; note that standard T-DM1 treatments are administered in three-week cycles, i.e on the same timescale [44]) than that of kinetics (minutes), i.e.

$$\min\{t_{\text{diff}}, t_{\text{ev}}, t_{\text{pd}}\} \gg \max\{t_{\text{on}}, t_{\text{off}}, t_{\text{int}}\}.$$

Thus we propose a quasi-static model reduction whereby the equation for $B$ is quasi-static (i.e. $\partial B/\partial t \approx 0$). This means that $B$ quasi-statically follows $C$ according to

$$B = \frac{k_{\text{on}}C_r}{\varepsilon(k_{\text{off}} + k_{\text{int}})}C. \qquad (4)$$

Substituting this result back into the equation for $C$, we obtain

$$\frac{\partial C}{\partial t} = D\nabla^2 C + \pi d_l P\left(C_{\text{plasma}} - \frac{C}{\varepsilon}\right)\delta_{\text{network}} - k_{\text{eff}}C, \qquad \text{where } k_{\text{eff}} = \frac{t_{\text{off}}}{t_{\text{on}}(t_{\text{off}} + t_{\text{int}})}$$

is the effective rate of the drug uptake. For the timescale of the effective uptake process we then have

$$t_{\text{eff}} = \frac{1}{k_{\text{eff}}} = t_{\text{on}}\left(1 + \frac{t_{\text{int}}}{t_{\text{off}}}\right) = 0.13 \text{ day} \ll \min\{t_{\text{diff}}, t_{\text{ev}}, t_{\text{pd}}\},$$

which holds true except for extreme combinations of very large local vessel diameter and very small inter-vessel distance. Thus, the distribution of the free drug on the timescale of interest (large times) follows from a linear quasi-static, which we refer to quasi-steady-state, reaction-diffusion equation

$$0 = D\nabla^2 C + \pi d_l P\left(C_{\text{plasma}} - \frac{C}{\varepsilon}\right)\delta_{\text{network}} - k_{\text{eff}} C, \qquad \text{where } k_{\text{eff}} = \frac{t_{\text{off}}}{t_{\text{on}}(t_{\text{off}} + t_{\text{int}})} \tag{5}$$

and $C_{\text{plasma}}$ is given in Eq (3). The bound-drug concentration $B$ then follows from the quasi-steady-state assumption in Eq (4) and the internalized drug $I$ is obtained simply by integrating the last equation in Eq (2). Detailed derivations pertaining to the proposed model reduction are summarized in S1 Text. Moreover, when posed on a simple geometry wherein a single cylindrical blood vessel perfuses the surrounding tissue, the reduced-order model is amenable to direct mathematical analysis - we found an exact solution to this problem and confirmed the validity of the model reduction via comparison with the numerical solution of the full problem — see Figs C–E in S1 Text. All dimensional parameters with appropriate units and references are summarized in Table A in S1 Text.

## Processing results

We define hypoxia as an oxygen concentration below 8 mmHg. We choose this value as it is defined as hypoxia in tumours [45], and because the efficacy of T-DM1 is reported to be reduced at that level of oxygenation [11]. Similarly, we determine that there is a sufficient concentration of T-DM1 when it is above the $\text{IC}_{50}$ (half maximum inhibitory concentration) reported in the literature, 2.9 nmol/L or 6.8 nmol/L, depending on the tumour cell line [11]. We report the free drug concentration to make it directly comparable to the $\text{IC}_{50}$ data available in the literature [11]. Therefore, there are two conditions required for optimal drug effect, which are a sufficiently high drug concentration, above the $\text{IC}_{50}$, and a sufficient oxygen concentration, above the hypoxia limit, leading to four possible conditions at any point in the tissue: 1) $C > \text{IC}_{50}$ and $C_{o_2} > C_{\text{hypoxia}}$, 2) $C > \text{IC}_{50}$ and $C_{o_2} < C_{\text{hypoxia}}$, 3) $C < \text{IC}_{50}$ and $C_{o_2} > C_{\text{hypoxia}}$, and 4) $C < \text{IC}_{50}$ and $C_{o_2} < C_{\text{hypoxia}}$.

The drug and oxygen concentrations are measured in two separate sections of the tissue being modelled. Firstly, the core of the tissue, which is defined as being within 1000 $\mu$m of the centre of the domain, which is the same region as the one where the blood vessels are compressed. The second, the periphery region, refers to the area of tissue wrapping around the core area, see Fig 2A–2C, and has the same surface area as the core region. We exclude tissue beyond that area due to boundary effects and avascular areas, see Fig A in S1 Text for details. The boundary effects result from the boundary conditions in the transport equations. The avascular areas arise due to the limits of two-dimensional network generation, leading to dead-end vessels with no flow in the network growth process.

The Student's t-test is used to test if the difference in mean of two distributions is statistically significant. The null hypothesis of the test is that the mean of the two distributions is the same. If the null hypothesis is rejected, that is the p-value from the test is below 0.05, the two distributions have a different mean.

Two methods are used to quantify how well the drug and oxygen concentrations overlap. Firstly, the overlap index is used [46],

$$O(x, y) = \frac{C(x, y)C_{O_2}(x, y)}{\max\{C\}\max\{C_{O_2}\}} \tag{6}$$

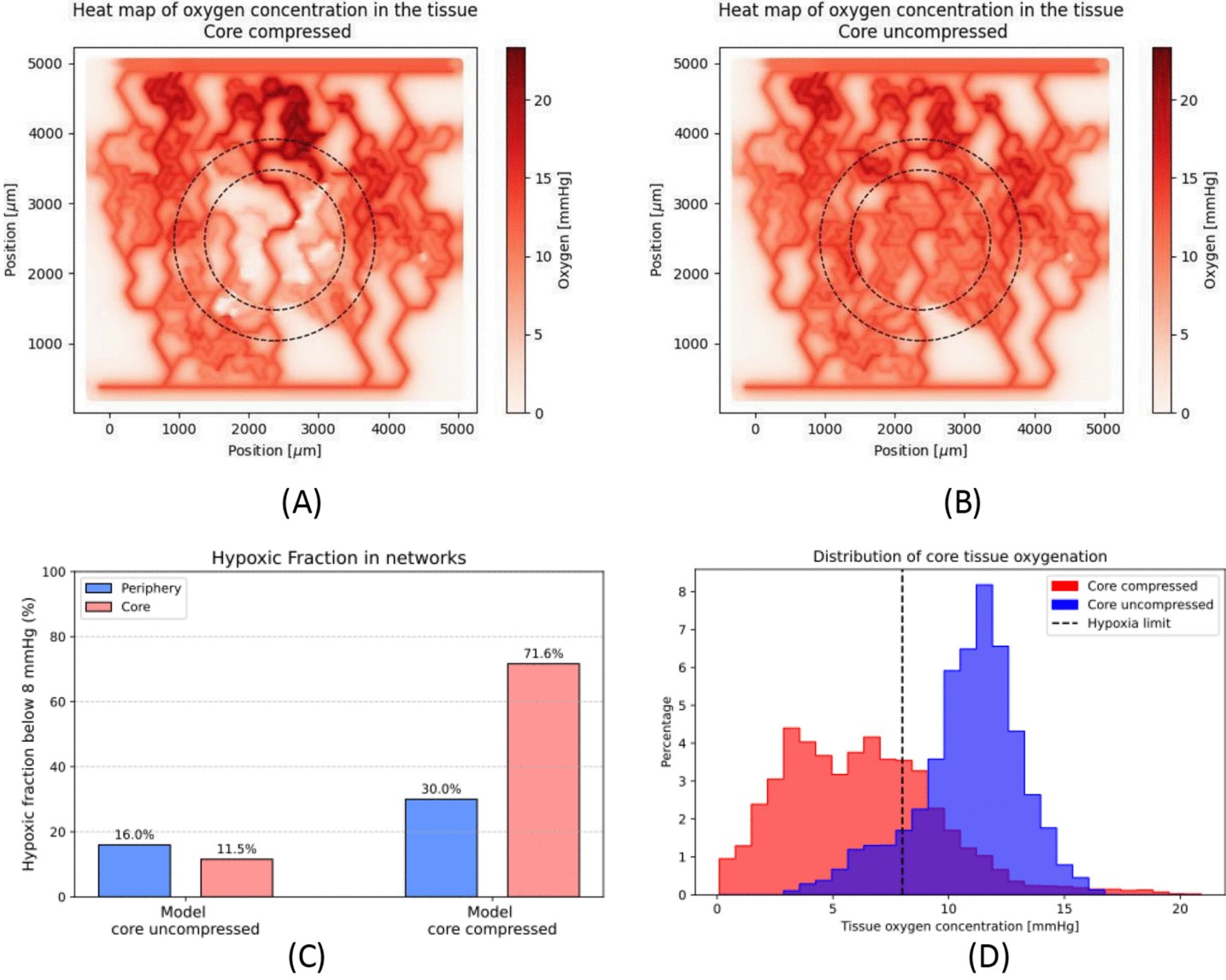

**Fig 2. (A) and (B) show tissue oxygenation in mmHg for the compressed model (A) and the uncompressed model (B), the dashed circle delineates the core, inner circle, and periphery, ring around inner circle, regions.** (C) shows the fraction of hypoxic tissue, in each model, in the core region. (D) shows histogram of oxygen concentration plots for each model in the core region, dashed line is the hypoxia limit at 8mmHg.

where $O(x, y)$ is the overlap index at position $(x,y)$, $C(x, y)$ and $C_{O_2}(x, y)$ are the T-DM1 and oxygen concentrations at position $(x, y)$, and $\max\{C\}$ and $\max\{C_{O_2}\}$ are the maximum T-DM1 and oxygen concentrations in the tissue. The overlap index is bound between 0 and 1, where 0 means at least one of the two is absent, whereas 1 indicates both are present in their maximum quantities. Secondly, we use Moran's Bivariate I to quantify the spatial correlation of oxygen with the drug, where Moran's bivariate I is the correlation coefficient of a variable in space with the weighted average of the other variable in the neighbouring positions [47]. A Moran's I of 0 indicates the two variables are uncorrelated, 1 that they are perfectly positively correlated, and -1 that they are perfectly negatively correlated [47].

## Results

### Compression reduces tissue oxygenation

We start by investigating the effect that the core vessel compression has on the fraction of hypoxic tissue. Fig 2A–2C shows heat maps for the oxygen concentration in the compressed and uncompressed model, respectively. The heat maps reveal that, in the compressed model, the core region has low oxygen concentration levels as well as some areas of low concentration in the periphery. On the contrary, in the uncompressed model, the core region has much higher oxygenation levels. In effect, in the core region the mean oxygen concentration rises from 6.24 mmHg (median 6.04 mmHg) in the compressed model to 10.85 mmHg (median 11.17 mmHg) in the uncompressed model, while in the core the compressed model has a more heterogeneous oxygenation with an interquartile range of [3.62 mmHg, 8.33 mmHg] compared to the uncompressed model interquartile range of [9.77 mmHg, 12.30 mmHg]. In addition to the increased heterogeneity, the oxygen concentration interquartile range in the core of the compressed model is entirely hypoxic (< 8 mmHg), contrary to the uncompressed model. The distribution of oxygen concentrations in the compressed model and uncompressed model are plotted in Fig 2D, showing a lower mean and wider distribution in the core of the compressed model compared to the core of the uncompressed model (difference is statistically significant).

We can observe, from Fig 2A–2B, that the periphery region also contains changes in the oxygenation between the two models. Fig 2C shows that the hypoxic fraction in the periphery of the core compressed model is higher than in the core uncompressed model, 30.0% compared to 11.5%. In addition, the change in hypoxic fraction between the core and periphery regions of the core uncompressed model is minimal, Fig 2C. The difference in hypoxic fraction in the core compressed model between the core and the periphery is attributable to network effects from the vascular compression [19].

We next interrogate what fraction of the core region is hypoxic, defined as having an oxygen concentration below 8 mmHg of oxygen. In the compressed model, the hypoxic fraction in the core area is 71.6%, falling to 11.5% in the uncompressed model. We hypothesise that the difference in hypoxic fraction between the compressed and the uncompressed models in the core region will reduce the efficacy of T-DM1, as T-DM1 is not as efficacious in hypoxic environments.

### Drug and oxygen overlap poorly and hypoxia reduces drug efficacy

Next, we investigate how the changed oxygenation of the tissue due to the tumour (compressed vessels) has an effect on the efficacy of T-DM1, using the model reduction we developed for T-DM1 transport (see Methods and Figs B–E in S1 Text). Fig 3A–3B shows binary heat maps indicating where there is both sufficient oxygen and sufficient T-DM1 to have a good drug effect. The heat maps clearly show an increased amount of the core tissue fraction having enough of both oxygen and T-DM1 in the uncompressed model compared to the compressed model, increasing from 28% to 82% of the tissue, Fig 3C–3D. We further investigate if that change in T-DM1 efficacy is changed by the $IC_{50}$ value used, and the results, Fig F in S1 Text, shows that the results with an $IC_{50}$ of 6.8 nM are similar to that of an $IC_{50}$ of 2.9 nM, Fig 3.

We then investigate what is the cause of the poor effect of T-DM1 in the tissue. Fig 3C–3D breaks down the effect of the drug in the tissue into its four components (with respect to sufficient oxygen and sufficient T-DM1, see methods for more details). It reveals that the loss in T-DM1 efficacy in the core is mostly due to the insufficient oxygen for the T-DM1 to internalise properly due to low oxygen concentration, with 3% of the tissue in the uncompressed case having sufficient T-DM1 but insufficient oxygen, compared to 57% in the compressed case.

Finally, we use two metrics to calculate the spatial correlation of oxygen concentration and T-DM1 concentration in the core, Moran's bivariate I and the overlap index, see methods for more details. In the compressed case, Moran's bivariate I is .27, increasing to 0.59 in the uncompressed case. While the overlap index increases from a mean of 0.09±0.12 in the compressed model to 0.18±0.21 in the uncompressed model, the difference is statistically significant. The increase in both Moran's I and the overlap index illustrates that the uncompressed model, through decompressed blood vessels, leads to an improved spatial correlation of oxygen concentration and drug concentration.

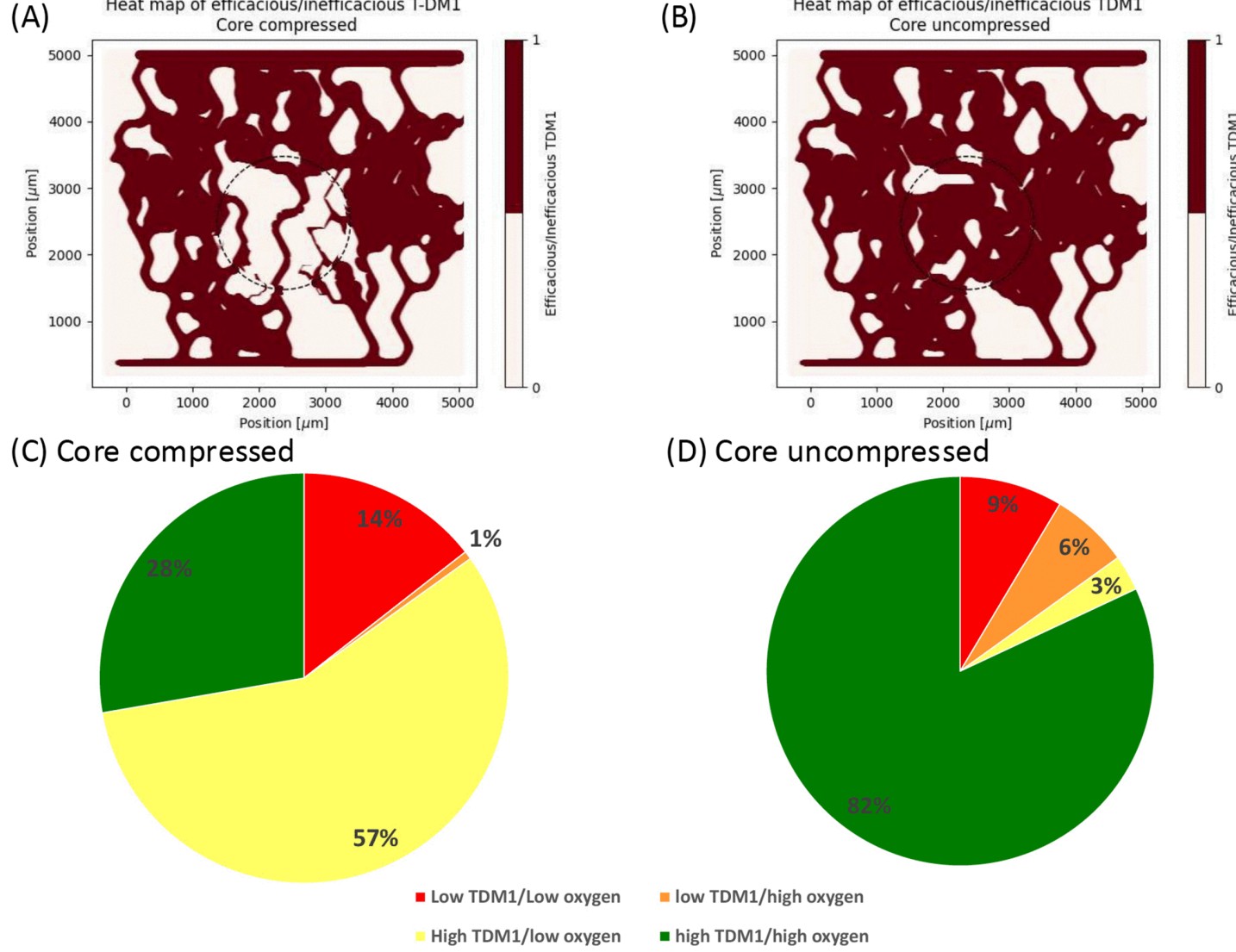

**Fig 3**. **(A) Shows efficacious T-DM1 (oxygen above 8mmHg and T-DM1 above 2.9 nM, both required for internalisation and killing cells, respectively) in the compressed simulation.** (B) Shows efficacious T-DM1 in uncompressed simulation. (C) Shows fraction of tissue in core region corresponding to sufficient/insufficient oxygen/T-DM1 in compressed (C) and uncompressed (D) cases.

## The effect of inlet haematocrit and T-DM1 depletion from blood stream

One might speculate that the extent of hypoxia can simply be alleviated via increasing the inlet discharge haematocrit which could in turn improve the efficacy of T-DM1. In order to assess its impact on T-DM1 efficacy, in Fig 4A–4B we vary the inlet discharge haematocrit between 5 and 30%. An inlet discharge haematocrit below 15% is too low to produce sufficient oxygenation even in the close promixity of the blood vessels, independent of the compression status, which results in almost the entire core tissue being hypoxic (Fig 4A). Above this threshold, increasing inlet discharge haematocrit indeed yields an improved drug efficacy. However, while in the uncompressed case the efficacy improves greatly so that for 25-30% inlet discharge haematocrit, the majority of the core tissue experiences sufficient concentrations of oxygen and T-DM1, the compressed model results in the majority of the core tissue remaining inefficaciously perfused even for 30%

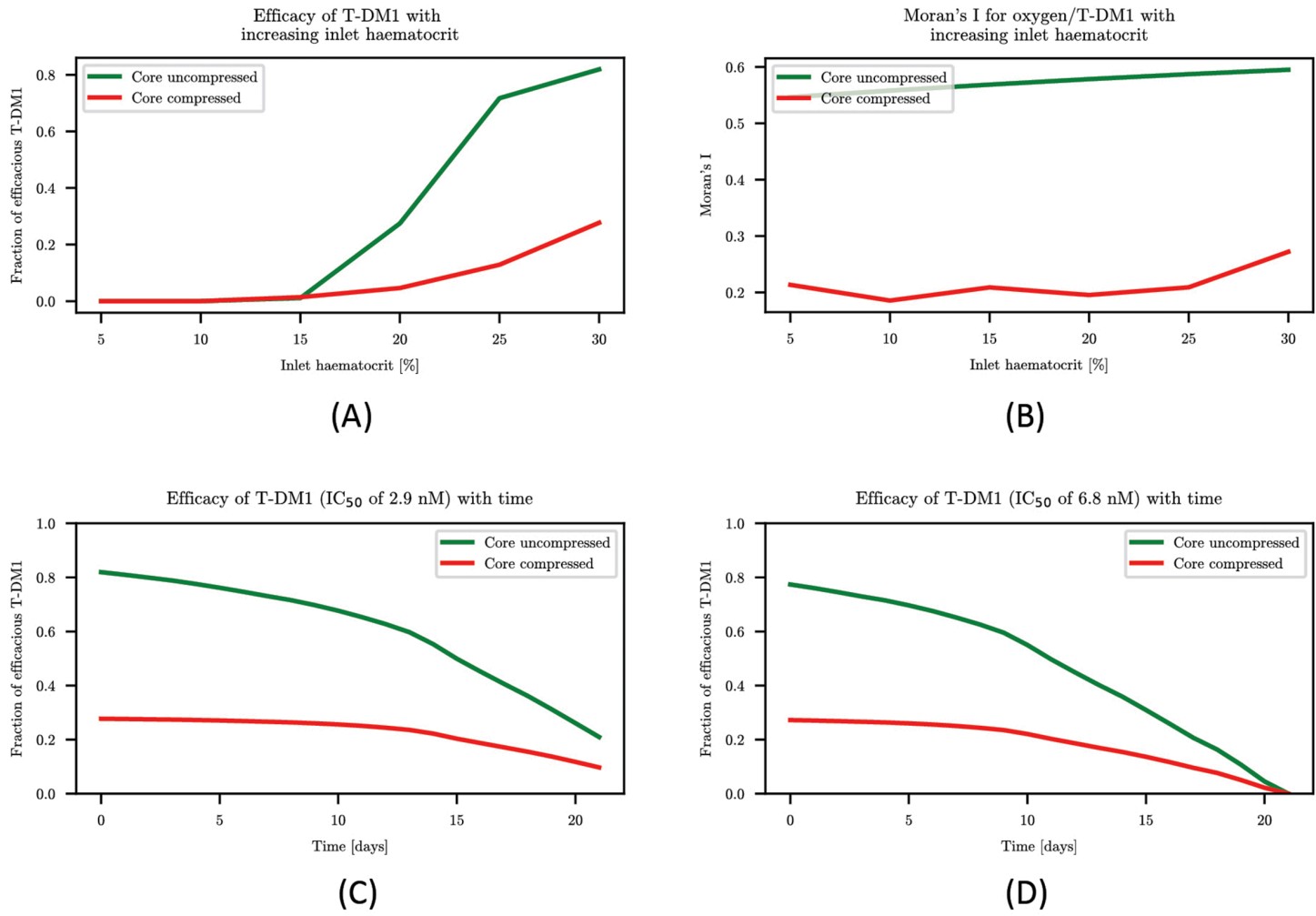

**Fig 4**. **(A) shows how increasing inlet discharge haematcorit changes the fraction of high oxygen/high T-DM1 in the tissue.** (B) shows how Moran's I changes with inlet discharge haematocrit. (C) shows efficacy of T-DM1 with time when $IC_{50}$ = 2.9 nM. (D) shows efficacy of T-DM1 with time when $IC_{50}$ = 6.8 nM.

inlet discharge haematocrit. The spatial correlation between oxygen and T-DM1, as evaluated by Moran's I, only varies mildly as inlet discharge haematocrit is increased independent of tumour status (Fig 4B). Therefore, the weak correlation observed when vessels are compressed cannot simply be rectified by increasing the inlet discharge haematocrit.

Motivated by the rate at which T-DM1 concentration in the blood stream decays, drug injections are standardly administered every 3 weeks [44] (see also Fig B in S1 Text). Fig 4C–4D documents that due to the decay of T-DM1 from plasma, the efficacy of drug in the core tissue decreases with time to a point where only a negligible fraction of the core tissue experiences efficacious T-DM1, using two thresholding values – $IC_{50}$ = 2.9 nM (panel c) and $IC_{50}$ = 6.8 nM (panel d) – for two cell lines studied in [11]. Interestingly, in the compressed model, the fraction of core tissue experiencing efficacious T-DM1 is low and essentially independent of the particular value of $IC_{50}$. Confirming again that the primary factor preventing efficacious treatment is the prevalence of hypoxia caused by the tumour induced vessel compression.

## Discussion

The tumour microenvironment acts as a barrier to drug delivery and leads to tumour tissue hypoxia. In addition, some drugs require the presence of oxygen to be fully functional. We therefore hypothesised that the TME leads to areas of high drug concentration poorly overlapping with areas of high oxygen concentration due to their different modes of transport to tissue. In this work we tested this hypothesis with a computational model of a tumour with compressed tumour blood vessels, with oxygen and T-DM1 (an oxygen-dependent drug) transport from the blood vessels to the tissue, compared to an uncompressed model.

We started by deriving a novel model reduction for T-DM1 transport based on key timescales, simplifying its numerical solution. We then simulated blood flow through tumour induced compression, and uncompressed, vessels which then acted as sources of both oxygen and T-DM1 to diffuse to the tissue. As the compressed vessels introduce haematocrit heterogeneity in the microvascular networks, the tumour compressed tissue oxygenation was found to be more heterogeneous than the uncompressed model, with a hypoxic fraction decreasing from 71.6% to 11.5% in the compressed region. We then showed that due to the higher fraction of hypoxia in the compressed model, there is a lower fraction of efficacious T-DM1, 28% compared to 82% in the uncompressed model, as its internalisation is impeded in hypoxic conditions. This reduction in efficacious T-DM1 is the result of a reduced overlap of areas of high oxygen concentration with areas of high T-DM1 concentration. Finally, we showed that increasing the inlet discharge haematocrit to the network is insufficient to improve the efficacy of T-DM1 to levels comparable to that of an uncompressed vascular network.

Our model makes assumptions and simplifications that we address here. Two-dimensional tissue transport simulations have been shown to overestimate hypoxia, while still capturing relevant trends, due to not accounting for vessels in neighbouring planes [48], as is also present in our two-dimensional network generated in Fig 1 which has avascular areas in two dimensions. In addition, we simplify the model for the T-DM1 drug transport to a quasi-steady-state model, where we assume that the timescale of the decay of the T-DM1 bolus injection is much slower than the reaction-diffusion of the drug in the tissue. This allows us to find a quasi-steady-state solution for the drug in the tissue for any fixed concentration in the blood stream, as if it were not decaying with time. We demonstrate that the quasi-steady-state model shows a good match with the numerical solution of the full model, provided that the quasi-steady-state model is not used for the early hours of the bolus injection. From 1 day onwards, the simplified model shows an excellent match with the full model, see Figs B–E in S1 Text. In addition, the simplified model does not work well for extreme combinations of large vessels and small distances between vessels, which are not present in our network. This assumption is only valid for the T-DM1 drug that we model, and would have to be verified for other drugs that have different transport timescales [5]. Finally, our work aims to study vascular compression independently of other tumour vascular abnormalities, so does not reproduce other barriers to transport existing in tumours [2].

One of the implications of this work is that independently considering oxygen transport and drug transport in oxygen dependent drugs can lead to lower drug efficacy than expected. Given that multiple important drugs, such as doxorubicin [12] and T-DM1 [11] amongst others [14], require the presence of oxygen to function, this work suggests that the overlap of drug and oxygen is an important biomarker for drug efficacy and therefore treatment outcome. We use tumour-induced vessel compression [31,49] and T-DM1 [44] as exemplars of a vascular abnormality and drug transport, and hypothesise that this effect would occur, and potentially be amplified, with other structural abnormalities; such as decreased interbifurcation length [18,21], increased disorder including deviation from Murray's law [50], tortuosity [51] and leakiness [52]; and other drugs with different transport properties [5].

Future work combining experimental validation and numerical modelling could enable one to predict what type of abnormalities present in a tumour microvascular network would lead to poor oxygen/drug overlap depending on the transport properties of a specific drug. One could then characterize the transport properties of microvascular networks via constructing governing dimensionless groupings which incorporate all key model parameters such as the vascular properties [53], the inlet discharge haematocrit as well as other parameters determining the oxygen and drug perfusion (see

[43] for the former and Table A in S1 Text for the latter). We expect that this approach would provide more robust insights into drug efficacy assessment (e.g. a phase diagram dividing the parameter space into subregions where drug or oxygen perfusion limits the treatment efficacy) independent of a particular network design.

Finally, normalisation therapy offers a means of therapeutically changing tumour microvascular networks, with the aim of temporarily making them more efficient [30,54]. For example, angiotensin inhibition decompresses vessels and improves tissue oxygenation [30], through the opening of blocked vessels [30] and haematocrit homogenisation [19]. However, despite its promising approach, normalisation therapy has remained challenging [55]. This work postulates that normalisation could aim to improve drug and oxygen overlap, which could be tested experimentally, to improve therapeutic efficiency.

## Supporting information

**S1 Text. Contains additional information on the blood flow model, the network generated for the simulations, and the drug transport model reduction, as well as the efficacious T-DM1 fraction at IC$_{50}$ of 6.8 nM.**
(PDF)

## Author contributions

**Conceptualization:** Romain Enjalbert, Jakub Köry, Timm Krüger, Miguel O. Bernabeu.

**Data curation:** Romain Enjalbert, Jakub Köry.

**Formal analysis:** Romain Enjalbert, Jakub Köry.

**Funding acquisition:** Miguel O. Bernabeu.

**Investigation:** Romain Enjalbert, Jakub Köry.

**Methodology:** Romain Enjalbert, Jakub Köry.

**Project administration:** Miguel O. Bernabeu.

**Software:** Romain Enjalbert, Jakub Köry.

**Supervision:** Timm Krüger, Miguel O. Bernabeu.

**Validation:** Romain Enjalbert, Jakub Köry.

**Visualization:** Romain Enjalbert, Jakub Köry.

**Writing – original draft:** Romain Enjalbert, Jakub Köry, Timm Krüger, Miguel O. Bernabeu.

**Writing – review & editing:** Romain Enjalbert, Jakub Köry, Timm Krüger, Miguel O. Bernabeu.

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
