## [Decision Letter · Decision Letter 0]

7 Apr 2025

PCOMPBIOL-D-25-00224

Abnormal vasculature reduces overlap between drugs and oxygen in a tumour computational model: implications for therapeutic efficacy

PLOS Computational Biology

Dear Dr. Bernabeu,

Thank you for submitting your manuscript to PLOS Computational Biology. After careful consideration, we feel that it has merit but does not fully meet PLOS Computational Biology's publication criteria as it currently stands. Therefore, we invite you to submit a revised version of the manuscript that addresses the points raised during the review process.

Please submit your revised manuscript within 60 days Jun 07 2025 11:59PM. If you will need more time than this to complete your revisions, please reply to this message or contact the journal office at ploscompbiol@plos.org. Please include the following items when submitting your revised manuscript:

We look forward to receiving your revised manuscript.

Kind regards,

Alison Marsden

Academic Editor

PLOS Computational Biology

Stacey Finley

Section Editor

PLOS Computational Biology

**Additional Editor Comments:**

While there is only one review, we believe it is constructive and fair and raises important points for the authors to address.

**Journal Requirements:**

At this stage, the following Authors/Authors require contributions: Romain Enjalbert, Jakub Köry, Timm Krüger, and Miguel Bernabeu. Please ensure that the full contributions of each author are acknowledged in the "Add/Edit/Remove Authors" section of our submission form.

5) We notice that your supplementary Figures, Tables, and information are included in the manuscript file. Please remove them and upload them with the file type 'Supporting Information'. Please ensure that each Supporting Information file has a legend listed in the manuscript after the references list.

Potential Copyright Issues:

i) Figure S1: We noted that you stated "Time evolution of plasma concentration using the highest-dose from [39] (reproduced from Figure 1a therein)" in the figure legend. It is licensed under CC BY-NC 2.0 license which is not compatible with our CC BY 4.0 license as it has additional restrictions. So, please remove or replace the figure as we will not be able to publish it.

7) Please amend your detailed Financial Disclosure statement. This is published with the article. It must therefore be completed in full sentences and contain the exact wording you wish to be published.

8) Your current Financial Disclosure states, " M.O.B. gratefully acknowledges funding from: Fondation Leducq Transatlantic Network of Excellence (17 CVD 03); EPSRC grant no. EP/X025705/1; British Heart Foundation and The Alan Turing Institute Cardiovascular Data Science Award (C-10180357); Diabetes UK (20/0006221); Fight for Sight (5137/5138); the SCONe projects funded by Chief Scientist Office, Edinburgh \& Lothians Health Foundation, Sight Scotland, the Royal College of Surgeons of Edinburgh, the RS Macdonald Charitable Trust, and Fight For Sight; the Neurii initiative which is a partnership among Eisai Co., Ltd, Gates Ventures, LifeArc and HDR UK."

However, your funding information on the submission form indicates one funder.  Please ensure that the funders match between the Financial Disclosure field and the Funding Information tab in the online submission form. Both locations should list the same funders, grant numbers, and recipients in the same order.

**Reviewers' comments:**

Reviewer's Responses to Questions

Reviewer #1: In this manuscript, the authors investigate the effect of oxygen overlap on anticancer drugs that require sufficient oxygen to be effective, such as T-DM1, using a computational model that simulates both compressed and uncompressed tumor vessels. The overall presentation and flow of the manuscript can be improved. Below are several detailed comments and suggestions for revision.

The introduction should include an overview of the tumor microenvironment (TME) and various normalization strategies. This addition would help readers understand how compressed vessels in tumors reduce oxygenation and impair the efficacy of oxygen-dependent anticancer drugs such as T-DM1. Moreover, the manuscript should describe normalization strategies that decompress the tumor vasculature, thereby improving oxygenation and drug efficacy.

A presentation of the most recent models that simulate tumor vasculature, along with a comparison to the current model, would clarify the need to simulate oxygen dynamics and the performance of oxygen-dependent drugs like T-DM1 simultaneously.

The manuscript currently presents both the control (healthy) and tumor vasculatures with similar abnormal structures. In other words, the control case represents decompressed tumor vessels, while the tumor case represents compressed tumor vessels. To avoid confusion, I suggest replacing the term “control” or “healthy” with “normalized-decompressed tumor vasculature” and using this definition consistently throughout the manuscript. This scenario could be discussed within the context of TME normalization and extracellular matrix (ECM) normalization.

Reordering Figures 2a and 2b so that the tumor’s compressed vasculature is presented first, followed by the normalized-decompressed vasculature, would improve the manuscript’s flow. Similarly, consider reordering Figures 3a and 3b.

Adding a comparison case featuring normal (well-organized) vasculature with decompressed vessels would enhance the manuscript’s results. This scenario can also be framed within the context of TME normalization, combining ECM and vascular normalization (see ref. "Combining two strategies to improve perfusion and drug delivery in solid tumors," DOI: 10.1158/0008-5472.CAN-13-1989).

The units for receptor concentration and drug concentration must be consistent for comparative purposes in Table S1 and Figures S2–S4. In instances where the drug-bound complex concentration approaches the receptor concentration, the kinetics described in Equation 2 may no longer be valid due to the assumption of a constant receptor concentration. The authors should discuss this issue and clarify whether the concentration of drug-bound complexes ever reaches that of the receptors. In Figure S4, only the internalized drug appears to reach that level.

In the “Processing results” subsection, the drug concentration is denoted as C_(T-DM1), whereas it was previously defined simply as C. Please use consistent terminology for all variables and parameters in both the manuscript and supplementary materials. Additionally, please explain why the overlap index is calculated based on the free drug rather than the bound or internalized drug. Based on the findings reported in (https://doi.org/10.1158/1541-7786.MCR-19-0856), hypoxia affects drug internalization. Thus, it might be appropriate to calculate the overlap index using the bound drug along with oxygen levels. The authors should provide calculations for both free and bound drug and include one set in the main manuscript and the other in the supplementary material.

For the statement, “The distribution of oxygen concentrations in the tumour model and control model are plotted in Figure 2d, showing a lower mean and wider distribution in the tumour model compared to the control model (difference is statistically significant),” the authors need to specify the statistical test used to compare the distributions and their means, as well as the significance level. They should also indicate whether these findings apply to both the core and the periphery of the tumor. In my opinion, the distributions in the periphery appear similar aside from a scaling factor; please review Figure 2d and revise accordingly. More detailed information is also needed for the statement regarding the overlap index, “While the overlap index falls from a mean of 0.18 ± 0.21 in the control case to 0.09 ± 0.12 in the tumour model, the difference is statistically significant.”

The model reduction for T-DM1 transport, along with Figures S2–S4, are not mentioned in the results section. This omission should be addressed.

The manuscript should clearly state where all quantified values are derived (e.g., the entire computational domain, the periphery, or the core). For instance, “As the tumour compressed vessels introduce haematocrit heterogeneity in the microvascular networks, the tumour tissue oxygenation was found to be more heterogeneous than the control, with a hypoxic fraction increasing from 11.5% to 71.6%. We then showed that due to the higher fraction of hypoxia in the tumour tissue model, there is a lower fraction of efficacious T-DM1 (28% compared to 82% in a control model) because internalization is impeded in hypoxic conditions,” it is not clear where these measurements were taken.

The authors should include a discussion of the limitations of the model and the underlying assumptions.

Although the authors have included a code availability statement, a data availability statement is missing. In accordance with the journal guidelines, all data necessary to replicate the study’s findings should be made publicly available.

Minor Correction:

Replace the highlighted text

“In addition, tumour 20 microvascular networks can have avascular areas that lead to hypoxic regions due to the limited diffusion 21 distance of oxygen”

with

“In addition, tumour microenvironment 20 can have avascular areas that lead to hypoxic regions due to the limited diffusion 21 distance of oxygen.”

**Have the authors made all data and (if applicable) computational code underlying the findings in their manuscript fully available?**

Reviewer #1: Yes

PLOS authors have the option to publish the peer review history of their article (what does this mean?). If published, this will include your full peer review and any attached files.

Reviewer #1: No

**Figure resubmission:**
---

## [Decision Letter · Decision Letter 1]

8 Aug 2025

PCOMPBIOL-D-25-00224R1

Abnormal vasculature reduces overlap between drugs and oxygen in a tumour computational model: implications for therapeutic efficacy

PLOS Computational Biology

Dear Dr. Bernabeu,

Thank you for submitting your manuscript to PLOS Computational Biology. After careful consideration, we feel that it has merit but does not fully meet PLOS Computational Biology's publication criteria as it currently stands. Therefore, we invite you to submit a revised version of the manuscript that addresses the points raised during the review process.

Please submit your revised manuscript within 60 days Oct 08 2025 11:59PM. If you will need more time than this to complete your revisions, please reply to this message or contact the journal office at ploscompbiol@plos.org. Please include the following items when submitting your revised manuscript:

We look forward to receiving your revised manuscript.

Kind regards,

Stacey D. Finley, Ph.D.

Section Editor

PLOS Computational Biology

**Additional Editor Comments:**

Although the reviewer from the first submission was not available, the manuscript has been evaluated by two independent reviewers. They raise issues regarding model details and assumptions. These must be clearly and satisfactorily addressed before the manuscript can be considered for publication. 

**Reviewers' comments:**

Reviewer's Responses to Questions

**Comments to the Authors:**

Reviewer #2: The authors have done a good job of addressing most of the comments from the original review(s) with the exception of some nomenclature issues. The nomenclature using ‘tumour’ and ‘tumour compressed’ is confusing. Aren’t all the results for a tumour. Please use the more accepted nomenclature suggested by the reviewer. Or perhaps call them ‘core uncompressed’ and ‘core compressed’. At places in the manuscript it seems like ‘tumour’ refers to only the core, other times to the entire domain.

Additional comments

The large avascular spaces in the vasculature that we see in the ‘periphery’ would only be expected in the core of a tumor where stresses due to cell proliferation have already compressed some vessels. So what difference does compression make?

While Tumorcode might apply Murray’s law, tumor vasculature does not satisfy Murray’s law. This approximation holds applies only to healthy tissues.

Figure 2 and the associated text do not make sense. Is Figure 2 mislabeled? Panels b and d show that the decompressed tumor has more hypoxia in the periphery than in the core. This is to be expected given that the avascular spaces in the periphery are much larger than in the core. However, panel c shows a much larger hypoxic fraction in the core than in the periphery.

The conclusions of the study seem obvious. Oxygen diffusion distances in most tissues are known to be about a hundred microns from the nearest well perfused vessel (as confirmed in the Figure 2). The authors provide analysis showing that a characteristic distance for the drug diffusion, sqrt(D/keff), is about 2 mm. We should expect any tissue with avascular spaces larger than 100 microns, but less than 2 mm should fall into the category of poor overlap (low oxygen/high drug). It would be helpful to include discussion of this ‘back of the envelope’ analysis.

Convective effects from highly permeable, tumour blood vessels creating high interstitial pressure in the tumour core may contribute to poor drug delivery. Please include discussion of this limitation.

More discussion of the limitations of using a 2-D model is warranted.

Reviewer #3: It has been shown that solid stress created by tumor cells can accumulate to the extent that they compress blood and lymphatic vessels. Here, the authors combine a model of blood flow/hemorheology with diffusion models for oxygen and a drug to analyze the effects of vessel compression on drug efficacy. They conclude that vessel compression creates regions where drug penetrates, but there is insufficient oxygen for uptake and killing.

The modeling is sound, and has some innovative aspects - particularly the inclusion of blood hemodynamics and partitioning in the network. The manuscript is well written, and the conclusions warranted.

However, there are concerns about the limited novelty, given the many other models and analyses in the literature. In addition, the model is perhaps overly simplistic and the interplay between hemodynamics and vessel compression geometry were not sufficiently explored.

Comments:

1. The model is solving a steady-state solution and assumes the tumor and vasculature are not changing over time. In reality, tumor cells would be dying due to the drug and lack of oxygen, thus changing metabolism locally. In addition, new vessels can sprout into hypoxic regions relatively quickly. Furthermore, the evolution of a tumor with high solid stress in the center – such as is modeled – would result in necrosis in the central region, with little metabolism or blood flow. So the initial condition of the simulations, with viable cells throughout the domain, is not justified. A time-resolved model that considers these processes would more closely reproduce tumor dynamics and give more relevant results.

2. The main innovation is the treatment of skimming and heterogeneity of hematocrit, but these aspects are not analyzed in detail. Since plasma skimming depends on the relative vessel diameters and branch angles, a sensitivity analysis should have been done with networks with various topologies/geometries. Also, a critical assumption in the model is the effect of solid stress on vessel shape. Uniformly deforming all vessels near the center to the same elliptical shape needs further justification, since vessel compression is expected to be more heterogeneous. Again, a sensitivity analysis should be shown.

3. The authors should discuss how the calculations of diffusion in 2D relate to 3D tissues, and whether this might affect the conclusions.

4. The assumption that the source term "strengths" are proportional to local vessel diameter needs further justification. Shouldn't this also depend on flow rates and concentrations?

5. Blood becomes more efficient at releasing oxygen when sO2 is low, but the authors apparently do not include accurate oxygen saturation dynamics. This is expected to have important effects on the predicted distributions.

6. The comments about vascular normalization need more clarification. Normalization depends largely on changes in vascular permeability, which changes transvascular plasma transport and affects interstitial flows (see, e.g. https://doi.org/10.1371/journal.pcbi.1011131). High permeability and plasma extravasation also affects hematocrit and hemodynamics. These mechanisms were not reproduced in the presented model.

**Have the authors made all data and (if applicable) computational code underlying the findings in their manuscript fully available?**

Reviewer #2: Yes

Reviewer #3: None

PLOS authors have the option to publish the peer review history of their article (what does this mean?). If published, this will include your full peer review and any attached files.

Reviewer #2: No

Reviewer #3: No

**Figure resubmission:**
---

## [Decision Letter · Decision Letter 2]

1 Dec 2025

Dear Dr Bernabeu,

We are pleased to inform you that your manuscript 'Abnormal vasculature reduces overlap between drugs and oxygen in a tumour computational model: implications for therapeutic efficacy' has been provisionally accepted for publication in PLOS Computational Biology.

Best regards,

Stacey Finley

Section Editor

PLOS Computational Biology

Reviewer's Responses to Questions

**Comments to the Authors:**

Reviewer #2: The authors have addressed my concerns. Thank you.

**Have the authors made all data and (if applicable) computational code underlying the findings in their manuscript fully available?**

Reviewer #2: Yes

PLOS authors have the option to publish the peer review history of their article (what does this mean?). If published, this will include your full peer review and any attached files.

Reviewer #2: No

---

## [Editor Report · Acceptance letter]

PCOMPBIOL-D-25-00224R2

Abnormal vasculature reduces overlap between drugs and oxygen in a tumour computational model: implications for therapeutic efficacy

Dear Dr Bernabeu,

I am pleased to inform you that your manuscript has been formally accepted for publication in PLOS Computational Biology. Your manuscript is now with our production department and you will be notified of the publication date in due course.

With kind regards,

Anita Estes
